# Impact of Covid-19 control strategies on health and GDP growth outcomes in 193 sovereign jurisdictions

Matt Boyd[1¤*], Michael G. Baker[2], Amanda Kvalsvig[2], Nick Wilson[2]

**1** Adapt Research Ltd, Reefton, New Zealand, **2** Department of Public Health, University of Otago Wellington, Wellington, New Zealand

¤ Current Address: Adapt Research Ltd, Reefton, New Zealand
* matt@adaptresearchwriting.com

## Abstract

The Covid-19 pandemic caused approximately 27.3 million excess deaths globally as of June 2024. Despite growing research on pandemic response factors, the effectiveness of different strategic approaches to Covid-19 control remains insufficiently investigated. We aimed to examine associations between Covid-19 pandemic control strategies (including stringent border restrictions) with age-standardized excess mortality and GDP per capita growth outcomes during 2020–2021. We analyzed 193 sovereign jurisdictions with existing Global Burden of Disease Study data. Jurisdictions were classified by implementation of exclusion/elimination strategies reported in published literature, and the level of border restriction measures based on the Oxford Stringency Index. Multivariable analyses adjusted for island status, GDP per capita, and an index of government corruption. Excess mortality was cube root transformed and GDP per capita log transformed for regression analysis. Jurisdictions implementing explicit exclusion/elimination strategies showed the lowest cumulative age-standardized excess mortality (-2.1/100,000) compared to others (166.5/100,000). Island jurisdictions experienced lower mortality (64.8/100,000) than non-islands (194.3/100,000). Duration of border restrictions correlated with reduced excess mortality in islands (Pearson's r = -0.624, p < 0.001; β -0.004, island interaction -0.005, p < 0.001), but not in non-islands. However, this effect weakened when controlling for government corruption in a subsample (lower corruption was associated with lowered mortality). No consistent significant relationships emerged between border measures and GDP growth, suggesting that stringent border restrictions in a pandemic may not significantly harm economies. We concluded that exclusion/elimination strategies and related stringent border restrictions were associated with better health outcomes, particularly for islands. Effectiveness was likely partially mediated by governance quality. Future pandemic planning should consider both control strategy selection and implementation context, both of which are modifiable.

**Data availability statement:** Study data is available here: https://adaptresearch.wordpress.com/wp-content/uploads/2025/07/250317-updated-master-uploaded-to-adapt-site.xls Alternatively it is available here: https://doi.org/10.17605/OSF.IO/WPTGY.

**Funding:** The authors received no specific funding for this work.

**Competing interests:** No support was received from any organisation for the submitted work beyond the authors primary affiliations; the authors declare no financial relationships with any organisations that might have an interest in the submitted work; and no other relationships or activities that could appear to have influenced the submitted work. MB is the owner and sole employee of Adapt Research Ltd, this does not alter our adherence to PLOS Global Health policies on sharing data and materials.

## Introduction

The Covid-19 pandemic has exacted an extraordinary toll on global health and economic systems. Current estimates indicate pandemic-related cumulative excess deaths of 27.3 million (95% uncertainty interval: 19.3 to 36.3) as of June 2024 [1]. The economic impact has been similarly profound, with the International Monetary Fund projecting total macroeconomic losses due to the pandemic at $13.8 trillion through 2024 [2].

Future pandemics are considered likely [3], with potential for catastrophic outcomes should engineered pathogens be involved [4]. This risk may be heightened by rapid developments in artificial intelligence that could facilitate the creation of such engineered pathogens [5,6].

Given the world now has five years of experience with managing Covid-19 it is important to reflect on which factors and responses were associated with differential Covid-19 pandemic outcomes. For example, research has found associations with trust in government [7], social cohesion [8], and pandemic preparedness [9].

In this paper we focus on investigating the role of strategic approaches to Covid-19 control, notably exclusion/elimination strategies compared with mitigation (including tighter suppression) as described in established strategic frameworks [10]. A major reason for focusing on strategies is that they are concerned with goals, rather than specific interventions such as the use of 'lockdowns'. When a lockdown is used for elimination it is acting as a 'circuit breaker' for a few weeks until there is evidence of zero transmission of Covid-19 in a jurisdiction, and such measures can then be relaxed without fear of an immediate resurgence. Lockdowns used for mitigation have a different goal of 'flattening the curve' so they generally need to be used with varying intensity for sustained periods, which is different from their use for elimination.

Several studies have examined the relationship between country-level strategic responses to the Covid-19 pandemic and their subsequent health and economic outcomes. A study published in December 2020 documented more favorable health outcomes and GDP projections among countries pursuing elimination strategies (China, Taiwan, Australia, and New Zealand) compared to European and North American nations implementing mitigation and suppression approaches [10]. Similarly, research analyzing 44 countries in 2021 found that five "elimination strategy countries" (Australia, China, Japan, New Zealand, and South Korea) experienced lower Covid-19 mortality rates and less severe effects on GDP growth than "suppression/mitigation strategy countries" [11].

Additional research published in 2021 corroborated these findings, reporting lower Covid-19 mortality rates among OECD countries utilizing elimination strategies (Australia, Iceland, Japan, New Zealand, and South Korea) versus those implementing mitigation strategies [12]. This study further concluded that "elimination is superior to mitigation for GDP growth on average and at almost all time periods" [12].

A 2023 study constructed a Covid-19 pandemic "shock index" incorporating health, behavioral, and economic indicators for 44 countries [13]. It determined that compared to "reactive" countries, "proactive" nations (Australia, New Zealand, and South Korea) "consistently outperform the others both in terms of robustness and

resilience," further noting that "proactive countries better preserve their performance over time whereas similarly performing reactive ones slide down." Most recently, a 2025 study reported that within high-income island nations that were in the OECD, those using exclusion/elimination strategies tended to have better mortality and economic outcomes than those using mitigation/suppression strategies [14].

Beyond economic and mortality metrics, mental health impacts have also been examined. A study of 15 countries with regular mental health surveys during 2020 and 2021 reported that "elimination strategies minimized transmission and deaths, while restricting mental health effects" [15]. The elimination strategy countries in this analysis included Australia, Japan, Singapore, and South Korea, though the study acknowledged that "changes in mental health measures during the first 15 months of the COVID-19 pandemic were small."

Research has also examined intra-country strategies. Canadian studies reported that "the Atlantic provinces (New Brunswick, Nova Scotia, Prince Edward Island, and Newfoundland and Labrador) and Northern Canada (Nunavut, Yukon and Northwest Territories) generally implemented a containment strategy" consistent with an "elimination" or "zero-COVID strategy" [16]. This approach demonstrated some success, with these regions reporting "long periods with no community cases" [17]. Martignoni et al. concluded that "While elimination may be a preferable strategy for regions with limited healthcare capacity, low travel volumes, and few ports of entry, mitigation may be more feasible in large urban areas with dense infrastructure, strong economies, and with high connectivity to other regions."

Border closures may be the most critical component of the elimination strategy, where it needs to be sustained for the duration of the response [18]. Hence it is useful to refer to this strategy as "exclusion/elimination". If implemented proactively early in a pandemic, stringent border restriction has potential to fully exclude a pandemic agent [19].

At the onset of the global Covid-19 pandemic, most Pacific Islands and Territories "implemented rapid border closures" [20] which can be considered an "exclusion strategy". These responses may have been generally effective given that one analysis has reported that Oceania had the lowest excess mortality rate of any region (for January 2020 to September 2022) [21]. Another analysis [22], found that Oceania (which included Australia in this work) had negative excess mortality in 2020 and a low positive excess in 2021.

Another study of excess mortality during the Covid-19 pandemic reported that this was lower for island nations [8]. Particular nations that were considered to perform relatively well include: Australia, Iceland, Singapore, New Zealand and Taiwan [23,24]. A robust and detailed 2025 study of New Zealand's excess mortality 2020–23 supports this finding for New Zealand [25]. Being an island nation also significantly increased the probability of a country pursuing an exclusion/elimination strategy [11].

But in terms of quantifying the impact of border closure, there have been only a few studies. One study "found no evidence in favor of international border closures" while finding "a strong association between national-level lockdowns and a reduced spread of SARS-CoV-2 cases." [26] While an analysis of 166 countries found that "total border closures banning non-essential travel from all countries and (to a lesser extent) targeted border closures banning travel from specific countries had some effect on temporarily slowing COVID-19 transmission in those countries that implemented them" [27].

## Aims and hypotheses

In light of the background above, we aimed to explore any associations between Covid-19 control strategies with both excess mortality and GDP growth outcomes. Specific hypotheses were as follows:

- Hypothesis One (strategic choices) was that the five jurisdictions that clearly used an exclusion/elimination strategy would have lower age-standardized excess mortality than other jurisdictions. We refer to this group as having 'explicit' exclusion/elimination strategies.

- Hypothesis Two (implementation of control measures) was that the jurisdictions using relatively fast and complete border closure (based on date of border closure and stringency of border restrictions) would have lower age-standardized

excess mortality than other jurisdictions (when adjusted for GDP per capita and island status). We refer to this group as having stringent border controls.

- Hypothesis Three was that both the above hypotheses would also apply to a relatively successful macroeconomic response in terms of GDP growth.

## Materials and methods

Ethical review not required for this analysis of jurisdiction-level data.

### Study jurisdictions

The study comprised 193 sovereign jurisdictions with excess mortality data for 2020–2021 from the Global Burden of Disease (GBD) study [28]. We used the super-region and region classifications from the GBD. Additional categorization was performed for island (n = 48) vs non-island status (n = 145).

We defined island jurisdictions in the dataset as those surrounded by water, while ignoring structural connections to other land masses such as bridges, causeways and tunnels (and so including Singapore, Bahrain and the UK as islands). In summary, the group of island jurisdictions included the following: individual sovereign islands, island archipelagos (e.g., Indonesia, Philippines), island continents (i.e., Australia), and where the jurisdiction had a land border with another jurisdiction on the same island (e.g., Ireland, UK, Timor-Leste, Papua New Guinea, Brunei, Dominican Republic, Haiti, Cyprus). We excluded South Korea (although it can be considered a pseudo-island) and jurisdictions with mixed characteristics but where the capital city was on a continental land mass (e.g., Malaysia, Corsica [France], Sardinia and Sicily [Italy]).

We excluded non-sovereign island states because it is not always clear what role the local government played in pandemic-related decision-making (as opposed to the colonial power). Namely: American Samoa, Bermuda, Cook Islands, Greenland, Guam, Niue, Northern Mariana Islands, Puerto Rico, Tokelau, Virgin Islands. We also excluded non-sovereign non-island jurisdictions (e.g., Hong Kong). We excluded North Korea as data seemed unreliable with conflicting reports.

### Identifying explicit exclusion/elimination strategy jurisdictions

As detailed in the *Introduction*, all the following jurisdictions have been described in at least one study as adopting an elimination strategy: Australia, China, Iceland, Japan, New Zealand, Singapore, South Korea, and Taiwan. But based on our own work relating to Iceland [29], we consider "mitigation/suppression" is more appropriate for that country. Also Japanese authors have explained why Japan did not use an elimination strategy [30]. Similarly, the South Korean Government never appeared to articulate an "elimination" or "zero-Covid" goal, and so we think a "mitigation/suppression" strategy is the most accurate description (albeit at a very high level of control during 2020). Therefore, our analysis defined just five 'explicit' exclusion/elimination strategy jurisdictions: Australia, China, New Zealand, Singapore, and Taiwan. Table 1 summarizes the key actions deployed by these jurisdictions to combat the pandemic.

### Identifying jurisdictions that used stringent border restrictions

For classifying border restrictions we considered the Oxford Covid-19 Government Response Tracker [43], specifically the date at which jurisdictions were coded to have reached border control stringency '4', meaning a ban on arrivals from all regions or total border closure (note this index does not report information pertaining to citizens of the jurisdiction who may have required levels of home or facility-based quarantine). We obtained this stringency data as published by Our World in Data [44], and filled data gaps where possible, expanding the dataset by eight additional jurisdictions based on analyst and media reporting and/or government documents. We specifically determined:

Global Public
Health
PLOS

**Table 1. Jurisdictions with explicit exclusion/elimination strategies and the main public health measures that they implemented in 2020–2021.**

| Jurisdiction | Border control measures | Other major public health control measures |
|---|---|---|
| Australia | Mandatory facility-based 14-day quarantine for all arriving travelers (albeit with multiple failures) [31]. | State Governments and the Commonwealth Government made extensive use of communication to increase knowledge and promote adherence with control measures. Particularly during times of outbreaks, the following were used at the state and locality levels: mass testing, contact tracing, quarantine and isolation, mandatory mask use, and lockdowns (including school and non-essential workplace closures). Mass vaccination began in February 2021. A Government Inquiry provides extensive details of the control measures taken [32]. |
| China | Mandatory quarantine for all travelers that included facility quarantine (14 days), followed by home quarantine (7 days), followed by further monitoring (7 days). | The Government made extensive use of communication to increase knowledge and promote adherence with "Zero-COVID" control measures. Internal travel restrictions, mass mask use and widespread digital surveillance were used. Particularly during times of outbreaks, the following were used: mass testing, contact tracing and digital surveillance, quarantine and isolation, and lockdowns (including school and workplace closures). Workers in high-risk sectors (e.g., aviation, hospitals, quarantine hotels) lived in tightly controlled "bubbles" to avoid community transmission. Mass vaccination began in mid-2021 with domestically produced vaccines. For reviews see: [33, 34] |
| New Zealand | Mandatory facility-based 14-day quarantine for all arriving travelers (albeit with multiple failures) [31, 35]. | The Government made extensive use of communication to increase knowledge and promote adherence with control measures. Particularly during times of outbreaks, the following were used: mass testing, contact tracing, quarantine and isolation, mandatory mask use, and lockdowns (including school and non-essential workplace closures). Mass vaccination began in February 2021. Compared to Australia, the control measures were classified as more proactive and also involving less time in lockdowns [36]. A Royal Commission of Inquiry provides extensive details of the control measures taken [37]. |
| Singapore | Mandatory stay-home notices (7- to 14-days) at home or dedicated facilities (depending on country of origin). Restrictions eased from August 2021 [38]. | The Government was relatively well-prepared given lessons learnt from SARS. It responded rapidly and used a "circuit-breaker" lockdown early in the pandemic [39]. Mass mask use and digital technologies were extensively used. Particularly during times of outbreaks, the following were used: mass testing, contact tracing and quarantine and isolation. Nevertheless, in 2020 Singapore was "barely able to control the spread in foreign workers' dormitories" [39]. Mass vaccination began in January 2021 with "rigorous public outreach and media coverage" [39]. |
| Taiwan | Used a mix of facility-based quarantine and home quarantine (albeit the latter with "digital fencing") [40]. | Taiwan had a particularly rigorous pandemic response with few outbreaks and no need for lockdown measures. Extensive public health infrastructure established pre-Covid-19 (as per Singapore), "enabled a fast coordinated response, particularly in the domains of early screening, effective methods for isolation/quarantine, digital technologies for identifying potential cases and mass mask use." [41] Masks were domestically produced. Taiwan also developed its own vaccines [42] and mass vaccination began in March 2021. For reviews see: [41, 42] |

- Time until closure (days from 1 January 2020 until level '4' implemented)

- Duration of closure (sum of days spent at level '4')

- Time until relaxing closure (days from 1 January 2020 until last level '4' day)

### Outcome data sources

**Excess mortality data.** Excess mortality and age-standardized excess mortality by year, for all the jurisdictions (2020 and 2021) was obtained from the GBD Study Demographics Collaborators (see Acknowledgements). The estimation methods for these data have previously been described in a GBD publication [28], and the age-standardized excess mortality variable used in our analysis represents the difference between observed all-cause deaths and expected deaths (based on pre-pandemic trends) during 2020–2021 combined, standardized across age groups to enable comparison between populations with different age structures.

We consider that excess mortality is an important measure of the true mortality impact from the Covid-19 pandemic given that it identifies the difference between observed all-cause mortality and mortality expected under normal conditions. Consequently, this measure is not subject to the high levels of under ascertainment seen with Covid-19 mortality surveillance [45].

**GDP per capita and GDP growth.** Given that excess mortality in 2020–2021 is associated with level of development (e.g., a sociodemographic index [46]), we treated GDP per capita as a covariate in the multivariable analyses. GDP data for these jurisdictions was sourced from the World Bank's World Development Indicators. The World Bank defines GDP per capita as GDP divided by midyear population. Specifically, we obtained the World Bank GDP per capita purchasing power parity (PPP) (constant 2017 international $) data, which included adjustment for inflation, i.e., the World Bank dataset "NY.GDP.PCAP.PP.KD". We then calculated GDP growth from 2019 to 2020, and from 2020 to 2021, defined as the percentage change in five-year geometric means. In some analyses, we report results for 'Low', 'Medium' and 'High' GDP jurisdictions based on tertiles of 2019 GDP.

**Control variables.** Our primary control variable was mean GDP per capita (as above) for the five-year period ending in 2019. We also obtained data from the Covid-19 NP Collaborators for the three variables they found consistently associated with Covid-19 outcomes [7]. These data represented the level of government corruption, trust in individuals, and trust in government. The data had been developed through principal components analysis of underlying global survey data, as described by the Covid-19 NP Collaborators in 2022 [7]. However, with the exception of government corruption, data was missing for the majority of island jurisdictions limiting potential inclusion. Therefore, the control variables were limited to GDP per capita along with a variable specifically representing the governmental corruption, and bureaucracy corruption, components of the World Values Survey, as developed by the Covid-19 NP Collaborators.

## Statistical analysis

We established Pearson's r correlations across relationships and performed multivariable linear regressions controlling for island status, GDP per capita and government corruption. To examine border restriction patterns and their associations with key outcomes, we categorized jurisdictions based on their implementation of level 4 border restrictions. After excluding jurisdictions with missing data, we created three groups: those that never reached level 4 border restrictions (0 days), those with a duration below the median number of days at level 4, and those above the median. We compared age-standardized excess mortality, GDP growth 2019–2020, and GDP growth 2020–2021 across these groups using one-way ANOVA. Mean values with standard deviations were calculated for each group. To account for geographical and jurisdictional characteristics, we conducted stratified analyses for islands, non-island jurisdictions, and specific regions (Oceania and Caribbean). For each stratum, we reported sample sizes, means with standard deviations, and p-values from ANOVA tests comparing the three level 4 restriction groups.

## Results

### Global analysis of excess mortality by GBD region and explicit exclusion/elimination status

Our analysis of 193 jurisdictions revealed substantial variation in Covid-19 mortality outcomes and control strategies across different geographic and socioeconomic contexts (Table 2). The overall age-standardized excess mortality for the 2020–2021 period was 162.1 per 100,000 population (SD = 154.4), with non-island jurisdictions experiencing significantly higher excess mortality rates (194.3) compared to island jurisdictions (64.8). Notably, explicit exclusion/elimination jurisdictions (n = 5), demonstrated negative excess mortality (-2.1), in stark contrast to non-exclusion/elimination jurisdictions (166.5) (Table 2, Fig 1).

Geographically, Sub-Saharan Africa exhibited the highest excess mortality among super-regions (311.7), while High-Income regions experienced substantially lower rates (27.8). Among the GBD regions, Southern Sub-Saharan Africa reported the highest mortality (601.1), whereas East Asia showed the lowest (-2.2).

**Table 2. Descriptive statistics by jurisdiction, exclusion/elimination strategy, and GBD region for the 2020 to 2021 period (note 'high-income' region consists of several high-income countries across different parts of the world, 'level 4' is the maximum Oxford Stringency Index border restriction).**

| Group | N | Cumulative age-standardized excess mortality (SD) per 100,000 population (2020–2021) | Number reaching level 4 border control (%)* | Median days at level 4 (IQR) |
|---|---|---|---|---|
| **Jurisdiction types** | | | | |
| All jurisdictions | 193 | 162.1 (154.4) | 159 (82.8%, n = 192) | 130.5 (59-224) |
| Non-island jurisdictions | 145 | 194.3 (158.9) | 120 (83.3%, n = 144) | 123.5 (59-207) |
| Island jurisdictions | 48 | 64.8 (84.8) | 39 (81.3%) | 160.5 (66.2-418) |
| Non-exclusion/elimination jurisdictions | 188 | 166.5 (154.1) | 156 (83.4%, n = 187) | 128.0 (60-221.5) |
| Explicit exclusion/elimination jurisdictions (see text) | 5 | -2.1 (7.5) | 3 (60.0%) | 200.0 (0-591) |
| **Super-region** | | | | |
| Central Europe, Eastern Europe, and Central Asia | 29 | 150.4 (77.7) | 25 (86.2%) | 86.0 (56-137) |
| High-Income | 35 | 27.8 (22.1) | 18 (51.4%) | 59.0 (0-268) |
| Latin America and Caribbean | 30 | 170.9 (135.0) | 25 (83.3%) | 143.0 (89.2-199) |
| North Africa and Middle East | 21 | 186.4 (108.9) | 18 (85.7%) | 138.0 (89-238) |
| South Asia | 5 | 161.7 (99.2) | 5 (100.0%) | 137.0 (61-163) |
| Southeast Asia, East Asia, and Oceania | 25 | 45.5 (56.4) | 22 (88.0%) | 348.0 (143-693) |
| Sub-Saharan Africa | 48 | 311.7 (175.5) | 46 (97.9%, n = 47) | 142.0 (106.5-208) |
| **Region** | | | | |
| Andean Latin America | 3 | 423.5 (143.6) | 3 (100.0%) | 199.0 (166-243.5) |
| Australasia | 2 | -6.4 (9.1) | 2 (100.0%) | 650.0 (620.5-679.5) |
| Caribbean | 16 | 114.0 (102.0) | 13 (81.3%) | 109.0 (66.2-183.2) |
| Central Asia | 9 | 175.2 (103.0) | 8 (88.9%) | 143.0 (124-406) |
| Central Europe | 13 | 140.5 (69.4) | 11 (84.6%) | 70.0 (21-84) |
| Central Latin America | 9 | 196.7 (99.4) | 7 (77.8%) | 153.0 (137-191) |
| Central Sub-Saharan Africa | 6 | 334.2 (89.0) | 5 (100.0%, n = 5) | 144.0 (110-156) |
| East Asia | 2 | -2.2 (5.6) | 1 (50.0%) | 100.0 (50-150) |
| Eastern Europe | 7 | 136.9 (55.3) | 6 (85.7%) | 107.0 (57.5-119) |
| Eastern Sub-Saharan Africa | 17 | 286.2 (161.1) | 16 (94.1%) | 121.0 (36-168) |
| High-Income Asia Pacific | 4 | 7.7 (9.1) | 2 (50.0%) | 188.0 (0-410) |
| High-Income North America | 2 | 69.0 (31.5) | 1 (50.0%) | 254.5 (127.2-381.8) |
| North Africa and Middle East | 21 | 186.4 (108.9) | 18 (85.7%) | 138.0 (89-238) |
| Oceania | 12 | 34.4 (65.4) | 12 (100.0%) | 768.0 (359.8-854) |
| South Asia | 5 | 161.7 (99.2) | 5 (100.0%) | 137.0 (61-163) |
| Southeast Asia | 11 | 66.4 (42.9) | 9 (81.8%) | 145.0 (83.5-278.5) |
| Southern Latin America | 3 | 53.5 (23.9) | 3 (100.0%) | 385.0 (373.5-394) |
| Southern Sub-Saharan Africa | 6 | 601.1 (205.2) | 6 (100.0%) | 216.0 (207.8-227.2) |
| Tropical Latin America | 2 | 132.0 (14.2) | 2 (100.0%) | 164.5 (144.2-184.8) |
| Western Europe | 24 | 27.4 (14.5) | 10 (41.7%) | 0.0 (0-87.5) |
| Western Sub-Saharan Africa | 19 | 236.1 (93.5) | 19 (100.0%) | 144.0 (114-202.5) |

Values are presented as mean (SD) for mortality, count (%) for level 4 incidence, and median (IQR) for days at level 4.

* Note that Equatorial Guinea lacked data for reaching 'level 4' border controls, hence some rows in this column are calculated with one less jurisdiction in the denominator.

**Fig 1. Cumulative age-standardized excess mortality (2020–2021).** By jurisdiction, exclusion/elimination strategy, and GBD super-region, with box plot of median value, upper and lower quartiles, and range (note 'high-income' region consists of several high-income countries across different parts of the world).

Most jurisdictions (82.8%) reached level 4 border restrictions, with a median duration of 130.5 days (IQR: 59–224). Western Europe stood out for having the lowest proportion of jurisdictions reaching level 4 restrictions (41.7%), while Oceania jurisdictions maintained the longest median duration at this level (768 days), highlighting diverse policy approaches and pandemic trajectories across regions.

### Global analysis of outcomes based on border restriction measures

Among jurisdictions that reached level 4 border restrictions, calculation of Pearson's r revealed no correlation between the number of days to enact level 4 and outcomes. However, there were strong negative correlations, particularly in island jurisdictions, between age-standardized excess mortality and duration of level 4 restrictions (p < 0.001), see Table 3.

Indeed, in this group the island interaction was highly statistically significant, when controlling for GDP per capita, and more than doubled the total effect of full border closure stringency on age-standardized excess mortality for islands (Table 4). Fig 2 illustrates the regression results for islands versus non-islands.

However, results of one-way ANOVA analyses stratified by jurisdiction type, region, and GDP per capita (tertiles), revealed a more mixed picture (Table 5). Jurisdictions that never implemented level 4 border restrictions often showed the lowest age-standardized excess mortality rates. For countries in the highest GDP tertile—especially non-island nations—GDP growth was more favorable in places that had "below median" duration of level 4 restrictions compared to those with either "above median" duration or no level 4 restrictions at all. Interestingly, Oceania showed a different pattern. There, GDP growth was actually higher in jurisdictions with "above median" duration of level 4 restrictions compared to those with "below median" duration (2.6% vs -1.1% for 2019–20, p = 0.004; and 1.2% vs -2.1% for 2020–21, p = 0.049). It's worth noting, however, that in Oceania, we didn't have any jurisdictions that completely avoided level 4 border restrictions for comparison.

### Analysis of outcomes restricted to island jurisdictions

When we took islands (n = 36) forward for regression analysis in isolation (based on the findings in Tables 3 and 4 that demonstrated strong correlation between excess mortality outcomes and duration of border measures for islands enacting Level 4 restrictions, but not for non-island jurisdictions), we found that the duration of level '4' border restrictions was

**Table 3. Correlations between border restrictions and outcomes by jurisdiction type (only for the 159 (83%) of jurisdictions that enacted the highest level of border restrictions, i.e., Oxford Stringency Index 'level 4').**

| Measure | Outcome | All jurisdictions | Islands | Non-islands |
|---|---|---|---|---|
| **Age-standardized excess mortality** | | | | |
| Days until level 4 | Age-standardized excess mortality | 0.031 (n = 159) | -0.256 (n = 39) | 0.112 (n = 120) |
| Days at level 4 | Age-standardized excess mortality | -0.473*** (n = 159) | -0.624*** (n = 39) | -0.127 (n = 120) |
| Days until relaxation | Age-standardized excess mortality | -0.363*** (n = 159) | -0.639*** (n = 39) | -0.031 (n = 120) |
| **GDP growth 2019–2020** | | | | |
| Days until level 4 | GDP growth 2019–2020 | 0.049 (n = 151) | -0.090 (n = 36) | 0.078 (n = 115) |
| Days at level 4 | GDP growth 2019–2020 | 0.056 (n = 151) | 0.294 (n = 36) | -0.069 (n = 115) |
| Days until relaxation | GDP growth 2019–2020 | 0.023 (n = 151) | 0.262 (n = 36) | -0.077 (n = 115) |
| **GDP growth 2020–2021** | | | | |
| Days until level 4 | GDP growth 2020–2021 | -0.101 (n = 150) | -0.072 (n = 36) | -0.114 (n = 114) |
| Days at level 4 | GDP growth 2020–2021 | -0.054 (n = 150) | 0.137 (n = 36) | -0.102 (n = 114) |
| Days until relaxation | GDP growth 2020–2021 | -0.116 (n = 150) | 0.100 (n = 36) | -0.173 (n = 114) |

**\*** p < 0.05, ** p < 0.01, *** p < 0.001; Values shown are Pearson correlation coefficients with significance levels asterisked and sample sizes (n).

**Table 4. Regression analysis of jurisdictions reaching level 4 border restrictions and outcomes (excess mortality and GDP growth).**

| Measure | Outcome | Main effect | Island interaction | N |
|---|---|---|---|---|
| **Age-standardized excess mortality** | | | | |
| Days until level 4 | Age-standardized EM | -0.002 | -0.012 | 151 |
| Days at level 4 | Age-standardized EM | -0.004*** | -0.005*** | 151 |
| Days until relaxation | Age-standardized EM | -0.003*** | -0.005*** | 151 |
| **GDP growth 2019–2020** | | | | |
| Days until level 4 | GDP growth 2019–2020 | 0.001 | -0.006 | 151 |
| Days at level 4 | GDP growth 2019–2020 | 0.001 | 0.004 | 151 |
| Days until relaxation | GDP growth 2019–2020 | 0.000 | 0.003 | 151 |
| **GDP growth 2020–2021** | | | | |
| Days until level 4 | GDP growth 2020–2021 | -0.004 | -0.000 | 150 |
| Days at level 4 | GDP growth 2020–2021 | 0.000 | 0.004 | 150 |
| Days until relaxation | GDP growth 2020–2021 | -0.001 | 0.003 | 150 |

* $p < 0.05$, ** $p < 0.01$, *** $p < 0.001$; EM = excess mortality; Main effect shows coefficient for border measure controlling for GDP and island status. Island interaction shows additional effect for island jurisdictions. N is the number of observations used in each regression model.

strongly associated with reduced age-standardized excess mortality in island jurisdictions (Table 6). In our most significant model ($R^2 = 0.576$, F = 22.42, $p < 0.001$), each additional day at level 4 restrictions was associated with a reduction in the cube root of age-standardized excess mortality (β = -0.006, $p < 0.001$), while controlling for log-transformed GDP per capita (β = -0.902, $p < 0.01$).

This model explained approximately 58% of the variance in mortality outcomes, which is substantial for a parsimonious model with only two predictors (border restriction duration and GDP per capita). While both variables made meaningful contributions, the higher statistical significance of the border measure duration ($p < 0.001$ versus $p < 0.01$ for GDP) suggests it was the more robust predictor. Notably, the models examining economic outcomes (GDP growth 2019–2020 and 2020–2021) showed no significant associations with any border measures, indicating that the mortality benefits of sustained border restrictions in islands were not offset by detectable macroeconomic costs in these jurisdictions during our study period.

However, when controlling for government corruption in a subsample of 19 island jurisdictions with available data, we observed a marked shift in results (Table 7). While the overall model fit improved substantially ($R^2 = 0.714$–0.731), the effect of days at level 4 border restrictions on lower mortality was no longer statistically significant (β = -0.002, $p > 0.05$). Instead, lower government corruption emerged as a significant predictor of lower age-standardized excess mortality (β = 0.697–0.843, $p < 0.05$). This finding suggests that the apparent protective effect of level '4' border restrictions may be partially confounded by governance quality (at least in this subsample of islands), with better-governed jurisdictions implementing more effective public health measures overall.

A stratified analysis (islands with corruption data [n = 19] vs those without such data [n = 17]) shows that both groups demonstrated similar relationships between duration of level 4 border measures and mortality (β = -0.004, $p < 0.05$; β = -0.006, $p < 0.001$, respectively). This finding suggests that selection bias in terms of which countries report corruption data is not driving the original findings. Consistent with most of our primary analysis, we found no significant associations between any border measures and economic outcomes when controlling for corruption.

### Description of jurisdictions with negative excess mortality

Of the seven jurisdictions with negative age-standardized excess mortality during 2020–2021 (Table 8), six were islands (Mongolia being the exception). Four of the seven implemented 'level 4' border closure and all used at least some facility quarantine (Taiwan used both types of quarantine and did high quality home quarantine with "digital fencing"). The earliest

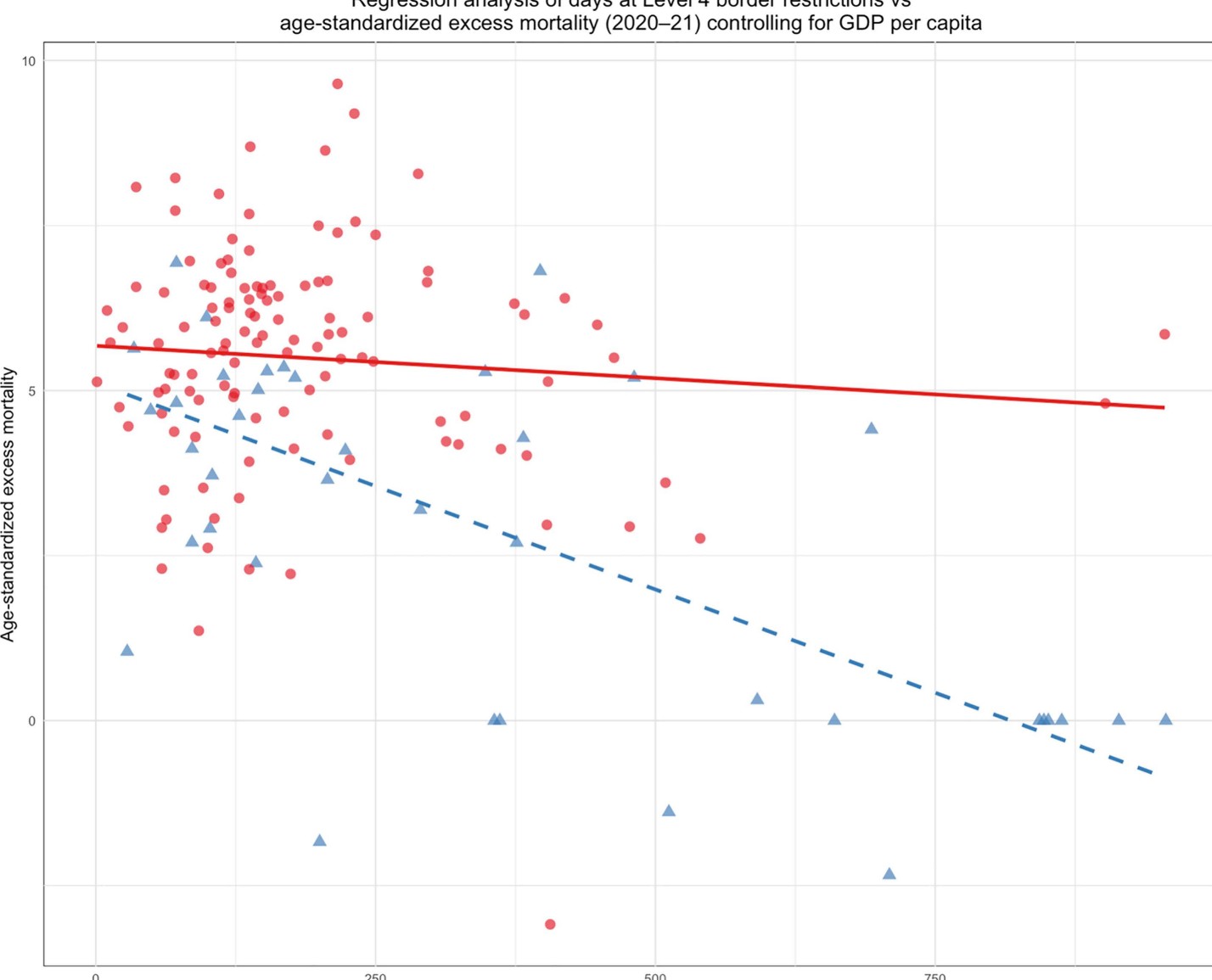

**Fig 2. Regression results for jurisdictions reaching level 4 border restrictions.** Showing duration of restrictions (days) vs age-standardized cumulative excess mortality for 2020-21 (cube root transformed) per 100,000 population; Outcome for non-islands (red) and island jurisdictions (blue). Note that cube root transformed excess mortality is sign preserving and adjusts for data skew; y-axis values of ~5 therefore represent excess mortality of ~125/100,000, and values of ~10 represent ~1,000/100,000.

end of cumulative negative excess mortality was in November 2020 in Iceland (which had incomplete border closure and mainly home quarantine). The mid-year for such a change was 2022 (for Barbados, Japan and Taiwan), and the last year was 2023 (for New Zealand, with Antigua and Barbuda still at negative excess mortality at the end of 2023). This group of jurisdictions included two of the five explicit exclusion/elimination jurisdictions (New Zealand and Taiwan).

**Table 5. Results of one-way ANOVA comparing border closure stringency category (i.e., no 'level 4' closure, below median duration, and above median duration of 'level 4' within each stratum) when stratified by island status, island region, and GDP per capita (tertiles).**

| Stratum | Outcome | N | No level 4 (n) | Below median (n) | Above median (n) | P-value |
|---|---|---|---|---|---|---|
| All jurisdictions | Age-standardized EM | 193 | 78.8±100.1 (33) | 188.7±138.9 (81) | 167.3±176.2 (78) | 0.002** |
| | GDP growth 2019–2020 | 182 | 0.6±2.3 (30) | 0.6±2.1 (79) | 0.2±3.0 (72) | 0.560 |
| | GDP growth 2020–2021 | 181 | 1.4±2.1 (30) | 1.0±2.4 (79) | 0.5±2.9 (71) | 0.259 |
| Islands | Age-standardized EM | 48 | 27.5±61.9 (9) | 102.8±84.0 (20) | 42.3±82.4 (19) | 0.025* |
| | GDP growth 2019–2020 | 45 | -0.3±3.0 (9) | -0.1±1.9 (17) | 0.6±3.0 (19) | 0.606 |
| | GDP growth 2020–2021 | 45 | 0.9±2.9 (9) | 0.1±1.9 (17) | 0.1±3.3 (19) | 0.752 |
| Non-islands | Age-standardized EM | 145 | 98.0±105.9 (24) | 203.9±147.9 (60) | 220.6±174.4 (60) | 0.004** |
| | GDP growth 2019–2020 | 137 | 1.1±1.9 (21) | 0.9±2.0 (59) | -0.1±3.0 (56) | 0.069 |
| | GDP growth 2020–2021 | 136 | 1.6±1.7 (21) | 1.3±2.3 (59) | 0.6±2.9 (55) | 0.168 |
| Oceania | Age-standardized EM | 12 | NA | 68.7±81.0 (6) | 0.0±0.0 (6) | 0.065 |
| | GDP growth 2019–2020 | 12 | NA | -1.1±1.0 (6) | 2.6±2.2 (6) | 0.004** |
| | GDP growth 2020–2021 | 12 | NA | -2.1±2.0 (6) | 1.2±3.0 (6) | 0.049* |
| Caribbean | Age-standardized EM | 16 | -7.4±54.8 (3) | 146.8±100.6 (7) | 136.4±83.0 (6) | 0.060 |
| | GDP growth 2019–2020 | 14 | -0.6±2.1 (3) | -1.5±2.1 (6) | -0.7±6.2 (5) | 0.937 |
| | GDP growth 2020–2021 | 14 | -0.9±2.0 (3) | -0.9±2.1 (6) | 0.6±7.1 (5) | 0.845 |
| Low GDP | Age-standardized EM | 61 | 228.0±141.2 (3) | 292.3±139.6 (29) | 198.7±187.3 (29) | 0.103 |
| | GDP growth 2019–2020 | 61 | 2.5±3.3 (3) | 0.8±2.1 (29) | 1.0±3.2 (29) | 0.584 |
| | GDP growth 2020–2021 | 61 | 2.4±2.3 (3) | 0.5±2.4 (29) | 0.8±3.0 (29) | 0.518 |
| Medium GDP | Age-standardized EM | 61 | 110.4±137.1 (9) | 149.2±98.1 (26) | 223.6±204.5 (25) | 0.107 |
| | GDP growth 2019–2020 | 61 | 0.1±3.2 (9) | 0.2±2.3 (26) | -0.1±3.2 (25) | 0.928 |
| | GDP growth 2020–2021 | 60 | 0.9±2.6 (9) | 0.8±2.5 (26) | 0.3±3.6 (24) | 0.818 |
| High GDP | Age-standardized EM | 60 | 41.1±38.3 (18) | 80.2±69.0 (21) | 71.7±73.7 (21) | 0.145 |
| | GDP growth 2019–2020 | 60 | 0.6±1.5 (18) | 1.2±1.6 (21) | -0.9±1.9 (21) | <0.001*** |
| | GDP growth 2020–2021 | 60 | 1.5±1.9 (18) | 2.0±1.7 (21) | 0.1±1.7 (21) | 0.003** |
| Low GDP Islands | Age-standardized EM | 15 | NA | 152.6±120.4 (8) | 12.2±32.4 (7) | 0.011* |
| | GDP growth 2019–2020 | 15 | NA | 0.7±3.3 (8) | 1.7±2.6 (7) | 0.567 |
| | GDP growth 2020–2021 | 15 | NA | 0.5±3.5 (8) | 0.7±2.8 (7) | 0.880 |
| Medium GDP Islands | Age-standardized EM | 15 | 5.1±51.2 (4) | 79.9±45.1 (6) | 84.2±77.4 (5) | 0.126 |
| | GDP growth 2019–2020 | 15 | -1.9±3.1 (4) | -1.1±2.4 (6) | 0.4±2.4 (5) | 0.437 |
| | GDP growth 2020–2021 | 15 | -0.6±1.8 (4) | -0.5±2.1 (6) | -0.7±3.5 (5) | 0.984 |
| High GDP Islands | Age-standardized EM | 15 | 45.4±69.1 (5) | 29.3±24.8 (5) | 28.9±63.4 (5) | 0.869 |
| | GDP growth 2019–2020 | 15 | 0.9±2.5 (5) | 0.4±1.4 (5) | -0.8±2.2 (5) | 0.423 |
| | GDP growth 2020–2021 | 15 | 2.1±3.3 (5) | 0.6±2.0 (5) | -0.3±1.9 (5) | 0.336 |
| Low GDP Non-islands | Age-standardized EM | 46 | 228.0±141.2 (3) | 320.6±142.6 (22) | 279.8±161.4 (21) | 0.496 |
| | GDP growth 2019–2020 | 46 | 2.5±3.3 (3) | 0.9±2.2 (22) | 0.6±3.1 (21) | 0.522 |
| | GDP growth 2020–2021 | 46 | 2.4±2.3 (3) | 0.6±2.6 (22) | 0.8±2.6 (21) | 0.538 |
| Medium GDP Non-islands | Age-standardized EM | 46 | 194.7±125.1 (5) | 174.7±103.5 (20) | 253.7±213.1 (20) | 0.314 |
| | GDP growth 2019–2020 | 46 | 1.7±2.5 (5) | 0.2±2.4 (20) | 0.3±3.3 (20) | 0.559 |
| | GDP growth 2020–2021 | 45 | 2.1±2.7 (5) | 0.8±2.8 (20) | 1.0±3.6 (19) | 0.706 |
| High GDP Non-islands | Age-standardized EM | 45 | 39.4±21.8 (13) | 87.4±62.0 (16) | 93.7±81.4 (16) | 0.053 |
| | GDP growth 2019–2020 | 45 | 0.5±1.1 (13) | 1.4±1.8 (16) | -0.8±1.9 (16) | 0.003** |
| | GDP growth 2020–2021 | 45 | 1.3±1.1 (13) | 2.2±1.8 (16) | 0.5±1.6 (16) | 0.014* |

Values are presented as mean±SD (n). EM=excess mortality; NA indicates insufficient data for analysis. GDP strata are divided into tertiles (Low, Medium, High) based on 2019 GDP values. Significance levels: * $p < 0.05$, ** $p < 0.01$, *** $p < 0.001$. Total N represents all observations available for that stratum and outcome.

**Table 6. Multivariable regression analysis for island jurisdictions comparing border restrictions and outcomes (excess mortality and GDP growth).**

| Outcome | Border measure | Border coefficient | GDP coefficient | R² | Adjusted R² | F-statistic | N |
|---|---|---|---|---|---|---|---|
| **Age-standardized excess mortality** | | | | | | | |
| Age-standardized EM | Days until level 4 | -0.013 | -0.446 | 0.118 | 0.065 | 2.22 | 36 |
| Age-standardized EM | Days at level 4 | -0.006*** | -0.902** | 0.576 | 0.550 | 22.42*** | 36 |
| Age-standardized EM | Days until relaxation | -0.006*** | -0.830* | 0.512 | 0.482 | 17.28*** | 36 |
| **GDP growth 2019–2020** | | | | | | | |
| GDP growth 2019–2020 | Days until level 4 | -0.003 | -0.462 | 0.042 | -0.016 | 0.72 | 36 |
| GDP growth 2019–2020 | Days at level 4 | 0.002 | -0.345 | 0.105 | 0.050 | 1.93 | 36 |
| GDP growth 2019–2020 | Days until relaxation | 0.002 | -0.377 | 0.090 | 0.035 | 1.64 | 36 |
| **GDP growth 2020–2021** | | | | | | | |
| GDP growth 2020–2021 | Days until level 4 | -0.003 | -0.191 | 0.011 | -0.049 | 0.18 | 36 |
| GDP growth 2020–2021 | Days at level 4 | 0.001 | -0.143 | 0.022 | -0.038 | 0.36 | 36 |
| GDP growth 2020–2021 | Days until relaxation | 0.001 | -0.169 | 0.014 | -0.046 | 0.24 | 36 |

* $p < 0.05$, ** $p < 0.01$, *** $p < 0.001$; EM = excess mortality; All models control for GDP (2019); Border coefficient represents the effect of the border measure; GDP coefficient represents the effect of 2019 GDP.

**Table 7. Multivariable regression analysis for island jurisdictions comparing border restrictions and outcomes (excess mortality and GDP growth) and including government corruption index as a control variable (n = 19 subsample of those islands with data).**

| Outcome | Border measure | Border coeff | GDP coeff | Corrupt coeff | R² | Adj R² | F-statistic | N |
|---|---|---|---|---|---|---|---|---|
| **Age-standardized excess mortality** | | | | | | | | |
| Age-standardized excess mortality | Days until level 4 | -0.009 | -0.919 | 0.843* | 0.731 | 0.678 | 13.62*** | 19 |
| Age-standardized excess mortality | Days at level 4 | -0.002 | -1.069 | 0.697 | 0.714 | 0.657 | 12.49*** | 19 |
| Age-standardized excess mortality | Days until relaxation | -0.002 | -1.118 | 0.703 | 0.717 | 0.661 | 12.67*** | 19 |
| **GDP growth 2019–2020** | | | | | | | | |
| GDP growth 2019–2020 | Days until level 4 | 0.000 | -0.038 | 0.291 | 0.023 | -0.173 | 0.12 | 19 |
| GDP growth 2019–2020 | Days at level 4 | 0.002 | 0.053 | 0.458 | 0.032 | -0.162 | 0.16 | 19 |
| GDP growth 2019–2020 | Days until relaxation | 0.001 | 0.086 | 0.455 | 0.033 | -0.160 | 0.17 | 19 |
| **GDP growth 2020–2021** | | | | | | | | |
| GDP growth 2020–2021 | Days until level 4 | -0.000 | 0.298 | 0.326 | 0.012 | -0.186 | 0.06 | 19 |
| GDP growth 2020–2021 | Days at level 4 | 0.001 | 0.384 | 0.498 | 0.020 | -0.176 | 0.10 | 19 |
| GDP growth 2020–2021 | Days until relaxation | 0.001 | 0.352 | 0.407 | 0.014 | -0.184 | 0.07 | 19 |

* $p < 0.05$, ** $p < 0.01$, *** $p < 0.001$; coeff = regression coefficient; All models control for GDP (2019) and government corruption; Border coefficient represents the effect of the border measure; GDP coefficient represents the effect of 2019 GDP; Corruption coefficient represents the effect of government corruption.

## Discussion

### Main findings and interpretation

Our analysis revealed several key patterns regarding the relationship between Covid-19 control strategies, border restrictions, and health and economic outcomes. The five jurisdictions implementing explicit exclusion/elimination strategies demonstrated the lowest mean age-standardized excess mortality, consistent with our first hypothesis. These jurisdictions also maintained the longest median duration of level 4 border restrictions, followed by island jurisdictions more generally, and then non-island jurisdictions.

Table 8. Case studies of jurisdictions with age-standardized negative excess mortality during the period 2020 to 2021.

| Jurisdiction (2021 Global Health Security Index score) | Border restriction type | Type of quarantine | Border restrictions | Further details (quarantine and easing of border controls) | When negative excess mortality ended[1] |
|---|---|---|---|---|---|
| Antigua and Barbuda (30.0) | Did not implement equivalent of level '4' Oxford Stringency Index border restrictions "screening, testing, monitoring and other measures" (https://antigua-barbuda.com/antigua-bar-buda-travel-advi-sory-as-of-14-octo-ber-2020) | Facility (including hotels) | Easing of border restrictions from June 2020, complete removal by August 2022 | Quarantine was "at an approved hotel" (https://en.wikipedia.org/wiki/COVID-19_pandemic_in_Antigua_and_Bar-buda). Regional travel bubble was used. The country reopened its borders to international travelers in a phased approach. | Still negative at the end of 2023 when available data on this ended |
| Barbados (34.9) | Did not implement level '4' | Facility | Used combination of level '1' and '2' from 26 Jan 2020 until Sept 2022. | Various quarantine facilities were described and: "arriving passengers will be subject to a 14-day quarantine" (https://en.wikipedia.org/wiki/COVID-19_pandemic_in_Barbados). May 2022 quarantine no longer required for unvaccinated travelers (https://www.gov.bb/news_article.php?id=118) Sept 2022 all travel protocols ended (https://www.totallybarbados.com/articles/health-fitness/health-care/covid-19/situation-report/) | March 2022 |
| Iceland (48.5) | Did not implement level '4' | Home (but also some facility) | Never really closed to EU tourists. Level '2' from 29 Jan 2020. Level '3' from 20 March 2020 until 30 June 2021. | Grout et al [29] note that Iceland used a mitigation strategy. Also that: "14-day quarantine policy was replaced by a new border screening program on 15 June 2020, which required test on arrival and self-quarantine while waiting for results." | November 2020 |
| Japan (60.5) | 512 days of level '4' | Mixture of facility and home quarantine, based on level of risk (e.g., unvaccinated and 'red' group countries highest risk) (https://h-crisis.niph.go.jp/archives/302438/) | Level '1' from 7 Jan 2020, Level '3' from 1 Feb 2020. Moved to level '4' border restrictions on 28 Dec 2020. | Japan has been described as having adopted a "mitigation" strategy [47]. Quarantine was only "until testing negative for COVID-19" (https://en.wikipedia.org/wiki/COVID-19_pandemic_in_Japan). On 11 October 2022, Japan reopened to non-citizens | March 2022 |
| Mongolia (41.0) | 406 days of level '4' | Facility | Level '3' from 27 Jan 2020, level '4' from 24 Mar 2020 until 3 May 2021. | Entry open from July 2021 with negative test and proof of vaccination (https://www.mongolian-ways.com/travel-blog/mongolia-coronavirus) | November 2021 |
| New Zealand (62.5) | 709 days of level '4' | Facility | Level '3' from 2 Feb 2020. Level '4' from 20 Mar 2020 until 26 Feb 2022. | New Zealand used both tight border control and an official elimination strategy [48]. It also used facility-based quarantine (which experienced failures [31, 35]). | April 2023 |

*(Continued)*

**Table 8.** (Continued)

| Jurisdiction (2021 Global Health Security Index score) | Border restriction type | Type of quarantine | Border restrictions | Further details (quarantine and easing of border controls) | When negative excess mortality ended[1] |
|---|---|---|---|---|---|
| Taiwan (NA) | 200 days of level '4' | Home (but also some facility) | Level '3' from 23 Jan 2020. Intermittent level '4' and '3' from 19 Mar 2020 until 31 Aug 2021. | The speed and rigor of Taiwan's response [49] indicated an exclusion and elimination strategy. "In October 2022, all quarantine requirements were removed" (https://en.wikipedia.org/wiki/COVID-19_pandemic_in_Taiwan). Of note was the high quality of home quarantine with "digital fencing" [40]. | June 2022 |

[1]Using cumulative excess mortality data available to the end of 2023 in Our World in Data.(1)

Our findings confirm previous research [8], showing that island jurisdictions experienced substantially lower excess mortality than non-island jurisdictions, with our results suggesting that relatively stringent border controls may have been a contributing mechanism. Additionally, we observed that some jurisdictions (n=7, of which six were islands, with high mean GDP per capita) achieved negative excess mortality. Several did so without implementing an explicit exclusion/elimination strategy (Table 8), although they all employed quarantine, regional travel bubbles, and/or level '3' or '4' border restrictions. These findings are consistent with our second hypothesis that effective implementation of stringent border restrictions was associated with lower excess mortality. We note that the absence of any islands in the Oceania region that did not enact 'level 4' border restrictions limited this study's ability to evaluate the impact of different levels of border stringency in the region.

Our analysis reveals a complex relationship between border control measures and pandemic mortality outcomes in island jurisdictions. Initial correlation results and regression models controlling for GDP per capita, demonstrated that longer border restrictions were significantly associated with reduced age-standardized excess mortality ($\beta$=-0.006, $p<0.001$ for days at level 4), suggesting strong protective effects. However, when controlling for government corruption (a methodological improvement that did come at a substantial cost to sample size), this relationship weakened considerably ($\beta$=-0.002, $p>0.05$), while lower government corruption itself emerged as a significant predictor of lower mortality outcomes ($\beta$=0.697, $p<0.05$). Importantly, a stratified analysis comparing jurisdictions with and without corruption data showed consistent negative associations between border measures and mortality in both islands that reported government corruption and those that did not ($\beta$=-0.004, $p<0.05$ and $\beta$=-0.006, $p<0.001$, respectively), indicating the fundamental relationship appears robust to selection bias.

These findings suggest that while relatively stringent border controls likely contributed to reduced pandemic mortality, their effectiveness cannot be fully separated from broader governance factors. The attenuation of border control effects when accounting for corruption indicates these variables share variance in explaining mortality outcomes, reflecting how effective governance may enhance implementation of public health measures. This nuanced picture aligns with emerging pandemic literature highlighting the multifaceted nature of successful responses, where specific interventions operate within broader systems of governance. Future research should continue to explore these complex interactions, as pandemic preparedness requires both specific policy interventions and the governance capacity to implement them effectively.

Regarding economic impacts, neither the original nor the corruption-controlled models revealed consistent relationships between border measures and GDP growth in general. Both sets of models exhibited weak explanatory power, with the corruption-controlled GDP growth models showing negative adjusted $R^2$ values that suggest overfitting. This pattern indicates that factors beyond those captured in our models likely played more important roles in determining economic

performance during the pandemic period. An important conclusion is that border restrictions in a pandemic may not significantly harm economies.

While this analysis found a strong association between use of a disease exclusion/elimination strategy and low excess mortality, it can only give us limited insights into the mechanism. In the early pandemic phase effective border measures are likely to have reduced mortality directly through preventing outbreaks in fully susceptible populations; genomic data from New Zealand show multiple introductions of the virus prior to implementation of border controls in 2020 [50], and a correspondingly high rate of border introductions in 2022 when border quarantine controls were lifted [51]. The delayed introduction of the virus allowed these jurisdictions to deliver vaccine to their populations and develop improved clinical management before widespread community transmission occurred. Vaccine was only widely available in 2021 which might explain why duration of border restrictions correlated with reduced excess mortality in islands. Some of the benefits of exclusion/elimination may only become apparent in subsequent years (e.g., 2022–2023) when Covid-19, particularly Omicron variants, began circulating widely even in previously protected populations. These additional benefits could include protection against long Covid where the risk is lower for people vaccinated prior to infection [52].

Our metric for speed of border restrictions (days from January 1, 2020, until level 4 implementation), while providing insights also highlighted methodological challenges. For example, New Zealand and Taiwan both implemented level '4' border restrictions at the same time (Table 8), but their response times relative to their first detected cases differed substantially (20 days for New Zealand versus 58 days for Taiwan). Conceptually, rapid border closure following the first detected case aligns closely with an exclusion/elimination approach. Ideally, border controls would be implemented before such cases arrived. These nuances, including time spent with less stringent border controls (level '3', etc) warrant further investigation in future research.

## Study strengths and limitations

Our study has several notable strengths, including the use of a comprehensive global dataset covering 193 jurisdictions with age-standardized excess mortality data. We employed a novel analytical approach examining the timing of "complete" border restrictions. Our multivariable analyses controlled for key confounding factors including GDP per capita and island status, while our exploration of government corruption as an additional control variable provided important insights into the role of governance quality during pandemic response.

However, several limitations should be acknowledged. Our analysis focused solely on sovereign jurisdictions, excluding sub-national entities that implemented successful exclusion/elimination strategies (e.g., Hong Kong [53], as did various Canadian Provinces [16]). Mean age-standardized excess mortality for the excluded non-sovereign islands (30.2/100,000) was lower than other islands, so this is likely a conservative exclusion, i.e., making it harder to find differences in excess mortality between islands and non-islands.

The most stringent level '4' of border closures may not accurately reflect actual enforcement, with potential discrepancies between reported policies and implementation, including possible circumvention via unofficial border crossings.

Additionally, our analysis including a government corruption metric was limited to jurisdictions with available data, potentially introducing selection bias and reducing statistical power. The Global Burden of Disease Study excess mortality estimates have inherent methodological limitations (e.g., for both mortality and demographic variables [28]). Furthermore, our analysis does not capture longer-term outcomes through subsequent years (e.g., 2022–2024). Important confounding factors such as trust in government, population health metrics (e.g., burden of chronic illness), and Covid-19 vaccination coverage in 2021 were also not included in our adjustments. Finally, economic impact assessments were complicated by the potential disproportionate effect on tourism-dependent economies and the potential economic benefits experienced by countries exporting information technology equipment and pharmaceutical and medical technology products during the pandemic.

This analysis is entirely based on the Covid-19 pandemic, a virus spread via the respiratory route and with specific, evolving infectious diseases dynamics. Consequently, our findings may have reduced generalizability to pandemics with very different characteristics, though the most probable future severe pandemic threats such as influenza share many of the same features [54]. It is notable that respiratory control measures used during the emergency phase of the Covid-19 pandemic eliminated transmission of a number of non-pandemic pathogens [55], including potential extinction of influenza B/Yamagata virus [56].

**Implications for policy and research**

Our findings suggest that while exclusion/elimination strategies and relatively stringent border controls may be associated with better health outcomes, particularly for island jurisdictions, the effectiveness of these approaches was likely partially mediated by broader governance factors including institutional quality and corruption control. This conclusion highlights the importance of considering not just what policies are implemented, but the quality and context of their implementation.

For future pandemic preparedness, our research suggests that border control strategies may be particularly effective for island jurisdictions, but their success likely depends on broader governance capabilities, resources, and institutional integrity. Simply implementing strict border measures without addressing underlying governance issues may not achieve optimal outcomes. Such measures could include the use of predetermined "green zones" (travel bubbles) and improved measures to prevent introduction of the infection by incoming travelers, enabling movement of people but not pathogens across borders.

Future research could attempt to better categorize pandemic response strategies of countries, notably distinguishing all of those that pursued exclusion/elimination (where this was an implicit as well as an explicit strategy) and the levels of control they achieved. It could further explore the complex relationships between governance quality, policy implementation, and pandemic outcomes using more comprehensive datasets that include measures of corruption and other social and governance metrics previously found to predict pandemic outcomes [7], for a wider range of jurisdictions. We were unable to meaningfully test many of these questions due to poor data coverage for islands.

Additionally, examining longer-term outcomes beyond our study period could provide valuable insights into the enduring impacts of different strategic approaches to pandemic control. Research could also assess how non-island countries such as Mongolia were also able to use, and in some cases sustain, exclusion/elimination strategies for prolonged periods without the border control advantages of island countries.

Global standard setting organizations such as the World Health Organization have an important role in providing a consistent strategic framework to support pandemic prevention, preparedness and response. As part of this role, they should investigate the use of standard terminology for pandemic response strategies as a key organizing concept [10]. Such a framework could also be integrated into the International Health Regulations and the process of responding to public health emergencies of international concern [57].

**Conclusions**

Our analysis of 193 jurisdictions provides insights into Covid-19 pandemic control strategies during 2020–2021. Jurisdictions implementing explicit exclusion/elimination strategies demonstrated substantially lower excess mortality than others, along with island nations with stringent border restrictions. While extended strict border control measures were associated with reduced mortality in island jurisdictions, this relationship weakened when controlling for government corruption, suggesting governance quality as a crucial mediator of policy effectiveness. Importantly, we found no consistent significant associations between border measures and GDP growth, challenging assumptions about inevitable health-economy trade-offs with strict border controls.

These findings suggest that exclusion/elimination may be the optimal pandemic control strategy for severe pandemics where it can be feasibly achieved. Circumstances supporting this response include island jurisdictions and strong

governance capabilities. Future pandemic preparedness should therefore consider both geographical context and governance quality when designing response strategies. Governance quality is modifiable. While geography is fixed, there are ways of achieving comparable border control attributes of islands with advanced planning. Such measures could include "green zones" and improved measures to prevent introduction of the infection by travelers. As the world prepares for future emerging infectious disease threats, these insights can inform more proactive approaches to pandemic prevention, preparedness and response.

## Acknowledgments

Prof Austin Schumacher from the GBD Collaboration for data sharing.

## Author contributions

**Conceptualization:** Matt Boyd, Michael G Baker, Amanda Kvalsvig, Nick Wilson.

**Data curation:** Matt Boyd, Nick Wilson.

**Formal analysis:** Matt Boyd.

**Investigation:** Matt Boyd, Nick Wilson.

**Methodology:** Matt Boyd, Amanda Kvalsvig, Nick Wilson.

**Project administration:** Matt Boyd.

**Resources:** Matt Boyd, Nick Wilson.

**Validation:** Matt Boyd, Nick Wilson.

**Visualization:** Matt Boyd.

**Writing – original draft:** Matt Boyd, Michael G Baker, Amanda Kvalsvig, Nick Wilson.

**Writing – review & editing:** Matt Boyd, Michael G Baker, Amanda Kvalsvig, Nick Wilson.

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
