## [Decision Letter · Decision Letter 0]

16 Jun 2025

PGPH-D-25-00821

Impact of Covid-19 Control Strategies on Health and GDP Growth Outcomes in 193 Sovereign Jurisdictions

Dear Dr. Boyd,

Thank you for submitting your manuscript to PLOS Global Public Health. After careful consideration, we feel that it has merit but does not fully meet PLOS Global Public Health’s publication criteria as it currently stands. Therefore, we invite you to submit a revised version of the manuscript that addresses the points raised during the review process.

We look forward to receiving your revised manuscript.

Kind regards,

Charin Modchang, Ph.D.

Academic Editor

Journal Requirements:

1. “Fig 1.tiff and Fig 2.tiff” files are over our file size limit of 10MB. This limit is in place for the convenience of reviewers, editors and readers. Please adapt this file so that the file size is below 10MB. You may find it helpful to consult our guidelines on compressing figures here: https://journals.plos.org/globalpublichealth/s/figures

Alternatively, you may wish to deposit large files in a separate repository to which you may link in your manuscript. You may find suggestions for repositories, including those which can accommodate large datasets, here: https://journals.plos.org/globalpublichealth/s/recommended-repositories 

Additional Editor Comments (if provided):

We have received comments from the reviewers. Please address them accordingly. Thank you.

Reviewers' comments:

Reviewer's Responses to Questions

**Comments to the Author**

1. Does this manuscript meet PLOS Global Public Health’s publication criteria ? Is the manuscript technically sound, and do the data support the conclusions? The manuscript must describe methodologically and ethically rigorous research with conclusions that are appropriately drawn based on the data presented.

Reviewer #1: Yes

Reviewer #2: Yes

2. Has the statistical analysis been performed appropriately and rigorously?

Reviewer #1: Yes

Reviewer #2: I don't know

3. Have the authors made all data underlying the findings in their manuscript fully available (please refer to the Data Availability Statement at the start of the manuscript PDF file)?

Reviewer #1: Yes

Reviewer #2: No

4. Is the manuscript presented in an intelligible fashion and written in standard English?

Reviewer #1: Yes

Reviewer #2: Yes

5. Review Comments to the Author

Reviewer #1: Overall comment:

Please include line numbers to easily reference issues in the paper next time.

The authors intended to evaluate the impact of Covid-19 control strategies on Health and GDP growth outcomes in 193 sovereign jurisdictions. The strengths of this paper is in the statistical analysis. It is well stipulated. The weakness of the paper is that you assume that everybody knows what you are talking about and have not defined the major terms in your study. You do not mention the elimination methods too.

Detailed comments:

How would you define age standardized excess mortality? How would you define GDP per capita and GDP growth?

Abstract: Looks good

Introduction: Looks good

Hypotheses: Look good

Methods and materials:

“We defined islands as per previous work” –we are reading this one and not your previous work. Please define it again and then you can reference your previous work.

Excess mortality data: Please mention the elimination methods used/ considered in this study.

Results: Well explained

Like I said earlier, we are working blindly with your terms. Define Low GDP, Middle and high GDP. What parameters were you looking at to define corruption in your jurisdictions for this study?

Discussion:

“Our analysis revealed several key patterns regarding the relationship between Covid- 19 control strategies, border restrictions, and health outcomes” – Is GDP a health outcome? Rephrase.

Reviewer #2: I should state that I am not a subject matter expert on this manuscript, I was able to read through with some observations:

1. The authors were able to comprehensively deal with the subject matter and provided references (sources) from which the data were extracted. However, there were no minimum data sets submitted.

2. Going by the calculations in Table 1 under the resuts, there appears to be some inconsistencies in the representation of percentages of the figures stated under ''Number reachng level 4 border control''. For instance, under All jurisdictions,a percentage of 82.8% was given which, i think should be 82.4% (82.38%); Non-island juridisctions given as 83.3% ought to be 82.8% (82.75%) ; Island juridisctions 81.2% ought to be 81.3% (81.25%) and Non-exclusion/elimination jurisdictions 83.4% ought to be 83% (82.97%). The same observations were made under super-region, Sub-Saharan Africa 97.9% which ought to be 95.8% Under region, Carribean given as 81.2% ought to be 81.3% (81.25%) while Central SubSaharan Africa given as 100% ought to be 83.3%. It is not clear how these figures will affect other tables.

6. PLOS authors have the option to publish the peer review history of their article (what does this mean? ). If published, this will include your full peer review and any attached files.

**Do you want your identity to be public for this peer review?** For information about this choice, including consent withdrawal, please see our Privacy Policy .

Reviewer #1: **Yes: ** Brian Asiimwe Kagurusi

Reviewer #2: No

---

## [Decision Letter · Decision Letter 1]

31 Aug 2025

Impact of Covid-19 Control Strategies on Health and GDP Growth Outcomes in 193 Sovereign Jurisdictions

PGPH-D-25-00821R1

Dear Dr Boyd,

We are pleased to inform you that your manuscript 'Impact of Covid-19 Control Strategies on Health and GDP Growth Outcomes in 193 Sovereign Jurisdictions' has been provisionally accepted for publication in PLOS Global Public Health.

Best regards,

Julia Robinson

Executive Editor

Reviewer #2:

Reviewer Comments (if any, and for reference):

Reviewer's Responses to Questions

**Comments to the Author**

1. If the authors have adequately addressed your comments raised in a previous round of review and you feel that this manuscript is now acceptable for publication, you may indicate that here to bypass the “Comments to the Author” section, enter your conflict of interest statement in the “Confidential to Editor” section, and submit your "Accept" recommendation.

Reviewer #2: All comments have been addressed

2. Does this manuscript meet PLOS Global Public Health’s publication criteria ? Is the manuscript technically sound, and do the data support the conclusions? The manuscript must describe methodologically and ethically rigorous research with conclusions that are appropriately drawn based on the data presented.

Reviewer #2: Yes

3. Has the statistical analysis been performed appropriately and rigorously?

Reviewer #2: Yes

4. Have the authors made all data underlying the findings in their manuscript fully available (please refer to the Data Availability Statement at the start of the manuscript PDF file)?

Reviewer #2: Yes

5. Is the manuscript presented in an intelligible fashion and written in standard English?

Reviewer #2: Yes

6. Review Comments to the Author

Reviewer #2: The manuscript has been refined to meet expectations.However, the manuscript is long with many tables. It is not clear at this end whether it will be more than the prescribed 10 MB. The manuscript is recommended for acceptance based on the technical/scientific contents.

7. PLOS authors have the option to publish the peer review history of their article (what does this mean? ). If published, this will include your full peer review and any attached files.

**Do you want your identity to be public for this peer review?** For information about this choice, including consent withdrawal, please see our Privacy Policy .

Reviewer #2: No
